# The Structure of Stable Cellulolytic Consortia Isolated from Natural Lignocellulosic Substrates

**DOI:** 10.3390/ijms231810779

**Published:** 2022-09-15

**Authors:** Grigory V. Gladkov, Anastasiia K. Kimeklis, Alexey M. Afonin, Tatiana O. Lisina, Olga V. Orlova, Tatiana S. Aksenova, Arina A. Kichko, Alexander G. Pinaev, Evgeny E. Andronov

**Affiliations:** 1All-Russian Research Institute of Agricultural Microbiology, 196608 Saint Petersburg, Russia; 2Department of Applied Ecology, Saint-Petersburg State University, 199034 Saint Petersburg, Russia; 3Dokuchaev Soil Science Institute, 119017 Moscow, Russia

**Keywords:** lignocellulose decomposition, microbial consortium, CAZymes, glycoside hydrolases, metagenome sequencing, amplicon sequencing

## Abstract

Recycling plant matter is one of the challenges facing humanity today and depends on efficient lignocellulose degradation. Although many bacterial strains from natural substrates demonstrate cellulolytic activities, the CAZymes (Carbohydrate-Active enZYmes) responsible for these activities are very diverse and usually distributed among different bacteria in one habitat. Thus, using microbial consortia can be a solution to rapid and effective decomposition of plant biomass. Four cellulolytic consortia were isolated from enrichment cultures from composting natural lignocellulosic substrates—oat straw, pine sawdust, and birch leaf litter. Enrichment cultures facilitated growth of similar, but not identical cellulose-decomposing bacteria from different substrates. Major components in all consortia were from Proteobacteria, Actinobacteriota and Bacteroidota, but some were specific for different substrates—Verrucomicrobiota and Myxococcota from straw, Planctomycetota from sawdust and Firmicutes from leaf litter. While most members of the consortia were involved in the lignocellulose degradation, some demonstrated additional metabolic activities. Consortia did not differ in the composition of CAZymes genes, but rather in axillary functions, such as ABC-transporters and two-component systems, usually taxon-specific and associated with CAZymes. Our findings show that enrichment cultures can provide reproducible cellulolytic consortia from various lignocellulosic substrates, the stability of which is ensured by tight microbial relations between its components.

## 1. Introduction

Plant biomass is one of the most abundant sources of organic carbon on the planet Earth, specifically due to the functioning of the modern industrial sectors [1]. Huge amounts of woody and herbaceous residues are generated as a byproduct of agriculture and forestry [2,3]. Most often these residues are considered waste; however, they can be reintegrated into the carbon cycle as a basis for the production of valuable products, such as biofuels, fertilizers, enzymes, organic acids, etc. [4].

The technical challenge of plant biomass handling lies in its chemical composition, the main component of which is lignocellulose [5], which, in turn, consists of three main compounds—cellulose, hemicellulose, and lignin, all highly recalcitrant to decomposition [6,7]. Other, less abundant compounds include proteins, pectin, soluble sugars, minerals, and lipids [8]. Their ratio differs between different lignocellulose-containing plant materials: cellulose makes up 9–80%, hemicellulose 10–50%, and lignin 5–35% [9]. The core of lignocellulose are cellulose microfibrils, surrounded by covalently linked molecules of hemicellulose and lignin. Cellulose is an unbranched homopolymer of glucose, while hemicellulose is a branched heteropolymer of different sugar residues, and lignin is an organic polymer derived from phenylpropane precursors [6,10].

Decomposition of lignocellulose is achieved by microorganisms using various multienzyme complexes. Cellulose processing is performed by three classes of enzymes: (1) endo-β-glucanases, which make cleavages in the internal bonds of the cellulose chain; (2) exo-β-glucanases, which cleave two (producing cellobiases) to four glucose residues from the cut in the chain; and (3) β-glucosidases (cellobiases), which convert cellobiose to glucose [11]. Hemicellulose decomposition requires additional specific enzymes, aimed at branching, acetylated sites, and different saccharide moieties, e.g., xylanases and mannases [12]. These enzymes belong to the group of glycoside hydrolases (GHs) which catalyze the hydrolysis of glycosidic bonds [13]. GHs work in conjunction with Carbohydrate-binding modules (CBMs), which are covalently attached to the catalytic domains of the enzymes and promote interaction of the enzyme with the substrate [14]. Variability of GHs and CBMs ensures affinity to different sources of lignocellulose [15,16]. Other enzymes, which take part in a cellulolytic complex, are glycoside transferases (GT), which catalyze the transfer of saccharide moieties [17]. They work with a variety of donors and both saccharide and non-saccharide acceptors; thus, their diversity is also very high [18]. Lignin is degraded by a spectre of oxidoreductases, including laccases, high redox potential ligninolytic peroxidases, and oxidases [19]. All these enzyme categories are cataloged as modules in the Carbohydrate-Active enZYmes (CAZy) Database, where they are classified into families [20]. These categories are highly diverse, e.g., GHs module consists of more than 100 families, defined more by sequence than by function [21]. Thus, enzymes specific for cellulose and hemicellulose decomposition are spread across many families of GHs. Enzymes for lignin decomposition are also present in CAZy’s Auxiliary Activities (AAs) module [22].

Both bacteria and fungi participate in the process of lignocellulose decomposition, using GHs of the same families [23,24]. Fungal enzyme systems are well studied, but unlike bacteria, they are harder to cultivate, and they do not tolerate extreme environmental conditions; thus, they are costlier to use industrially [25]. Bacteria, on the other hand, are widely distributed in aerobic, anaerobic, saline, and extreme temperature habitats, and therefore can serve as a source of universal biodegraders. The most active members of cellulolytic community are found in Actinobacteria, Proteobacteria, Firmicutes, Chloroflexi, and Bacteroidetes phyla [26].

There are works focused on isolating cellulolytic bacterial strains from various biomass-containing environments, such as soils or plant and manure compost [27,28,29]. All these studies propose isolated strains as candidates for industrial lignocellulose conversion. However, in natural ecosystems’ degradation of lignocellulose, biomass is carried out by multiple microorganisms with highly diverse spectres of enzymes, adaptive to different environmental conditions [30,31]. Thus, using bacterial consortia for biomass decomposition can overcome such difficulties in handling one-strain systems, such as negative feedback regulation and metabolite repression [32,33]. A way to isolate microbial consortium is by making enrichment cultures from natural communities using a specific carbon source [26,34,35]. This way the complex community of a natural ecosystem is funneled into simplified (with less variety) functional consortium specialized in this source [36]. Cortes-Tolalpa showed that microbial consortia derived from three different natural forest sources and enriched by wheat straw develop similar lignocellulolytic functional patterns while retaining unique taxonomic imprints [30]. However, not a lot of studies are focused on selecting reproducing minimal stable microbial consortia, which could be effectively saved and stored without altering their features.

In our study we aimed to isolate and explore stable cellulolytic consortia from enrichment cultures from composting plant biomass, common for northern regions of Europe-oat (*Avena*) straw, pine (*Pinus*) sawdust, and birch (*Betula*) leaf litter. These substrates represent different types of lignocellulose biomass—waste biomass (byproducts of agriculture and forestry) and virgin biomass (parts of naturally occurring plants), so our experiment aimed at incorporating different sources of common plant residues and testing their lignocellulolytic potential. We assessed the taxonomic structure and functional profiles of the consortia derived from these substrates using methods of high-throughput sequencing of 16S rRNA gene amplicons and full metagenome analysis.

## 2. Results

As a result of the experiment on enrichment cultures from three different composting plant biomass sources, we acquired four reproducible microbial consortia with different phenotypic features, such as coloration of the filter paper and gas emissions (Appendix A). All consortia visually macerated filter paper (Figure 1). The specifications of the consortia are presented in Table 1. They covered all cellulose-containing substrates used in the experiment and represented some of the composting periods: consortia OS2 and OS4 from oat straw (2 and 4 months), consortium PS4 from pine sawdust (4 months), and consortium BL6 from birch leaf litter (6 months). Other combinations of substrates and composting periods did not result in reproducible consortia. Different composting substrates required different dilutions for the acquisition of minimal cellulolytic microbial cultures.

We aimed to address the microbial composition of isolated consortia by sequencing 16S rRNA gene and ITS amplicon libraries. Eukaryotes in all consortia were identified (both from amplicon and metagenome sequencing data) only as a minor component. The relative abundance of eukaryotes in the consortia according to metagenomic data was only 0.2%. For consortia OS2 and BL6, the presence of ciliates was shown (Appendix A). Consortia OS4 and BL6 had some proportion of unidentified fungi. Based on the obtained data, we can state that the isolated consortia are predominantly bacterial.

### 2.1. Taxonomic Analysis of Cellulolytic Consortia by Amplicon Sequencing

#### 2.1.1. Alpha and Beta-Diversity of Consortia

As a result of amplicon sequencing, we acquired 24 libraries (six replicates for each of the four consortia) of 16S rRNA gene amplicons totaling 903,148 reads, which were divided into 535 phylotypes. All consortia demonstrated significant differences across the three indices of alpha-diversity (Figure 2). The highest values of all alpha-diversity indices were detected for the OS2 consortium, which was isolated from the substrate with the shortest composting period. Consortia OS4 and PS4, which were isolated from the substrates with 4-month composting periods, did not show significant differences in Simpson and Inverted Simpson indexes. Consortia BL6 had higher values of these indices than OS4 and PS4, but lower than OS2.

According to the beta-diversity metrics of 16s rRNA sequencing data, all four consortia were significantly separated from each other (R^2^ = 0.7, *p*-value < 0.001) (Figure 3), even OS2 and OS4 from the same substrate (R^2^ = 0.85, *p*-value = 0.0018). However, distances between consortia from oat straw were shorter than between others from different substrates. Bray–Curtis, UniFrac, and Weighted UniFrac metric systems showed the same trend but revealed differences in varying degrees (Appendix A).

#### 2.1.2. Taxonomic Composition of Consortia

According to amplicon sequencing, the main components of all consortia belonged to Proteobacteria and Bacteroidota phyla (Appendix A). To select the minimal cellulolytic community, consortia cultivated from the maximum dilution of the substrate, which retained the ability to decompose the paper filter, were taken into the study. Despite this, it turned out that each consortium consists of around a hundred phylotypes. Almost all of them were unique to different substrates, and only three minor phylotypes belonging to *Xanthobacter*, *Pseudoxanthomonas* and *Dokdonella*) were common for all four consortia (Appendix A). However, phylotypes pooled on the genus level revealed overlapping representatives. Major genera can be seen on the heatmap with relative abundance data (Figure 4). *Pseudoxanhtomonas* and *Devosia* genera from Proteobacteria were a major part in all consortia. Other phylotypes were distributed more differentially: *Sporocytophaga* (Bacteroidota) in OS2 and PS4; *Asticcacaulis* (Proteobacteria) in OS2, OS4, and PS4; *Cellulomonas* (Actinobacteroidota) in OS2, OS4, and BL6; *Caenimonas* and *Cellvibrionaceae* (Proteobacteria) in OS4; *Flavobacterium* (Bacteroidota) and *Blrii41* (Myxococcota) in OS2 and OS4; *Paenibacillus* (Firmicutes) and *Methyloversatillis* (Proteobacteria) in BL6; *Cohnella* (Firmicutes) in PS4 and BL6; *Pirellula* (Planctomycetota) in PS4.

To overcome the limitations of relative abundance analysis, we compared phylotype composition between consortia by compositional PhILR-transformation analysis, which reveals associated shifts in taxon composition between each pair.

Consortia OS2 and OS4 from an oat straw substrate shared the most phylotypes, despite having different phenotypes: the number of the common phylotypes was 65, 68.4% of the total read count from these consortia (Appendix A). Only 4% of the total reads were unique for OS4 in comparison with OS2, which had 26.8% of unique reads. The most abundant phylotypes from both consortia were similar, and consisted of representatives of Bacteroidetes, Gammaproteobacteria, Alphaproteobacteria, Myxococcota, Actinobacteria, and Verrucomicrobia. The main differences were at the low taxonomic level in members of Bacteroidota (Sphingobacteriaceae, between representatives of the genus *Parapedobacter*) and Proteobacteriota phyla (between phylotype groups within Comamonadaceae, Devosiaceae, Xanthomonadaceae, and *Bosea*). Combined data demonstrate that OS4 consortium differs from OS2 in reduced phylotype diversity and probably represents a microbial community with an increased specialization in cellulose degradation. Based on this conclusion, other comparisons were made only between consortia OS4, PS4, and BL6.

The comparison of the consortia from sawdust and leaf litter (PS4 and BL6) with those from straw (OS2 and OS4), revealed differences at higher taxonomic levels. Consortia OS4 and PS4 shared 29.6% of the total read count (Appendix A), OS4 and BL6—33.3% (Appendix A), PS4 and BL6—16.4% (Appendix A). In the PS4 consortium, the main changes occured within Alphaproteobacteria. The representation of Rhizobiales was increased, but at the same time, the representation of the *Devosia* genus decreased. The presence of Burkholderiales decreased, but Alcaligenaceae (*Achromobacter*, *Pigmentiphaga*) increased. In the order Caulobacterales, the composition of phylotypes within the genus *Asticcacaulis* changed. Among other phyla, a decrease in the diversity of Verrucomicrobiota with an increase in phylotypes of the genus *Opitutus* was noted.

The BL6 consortium from leaf litter differed most from the others in the presence of a large number of representatives of the Firmicutes phylum, which were underrepresented in other consortia. The Alphaproteobacteria group was characterized by the increase in *Magnetospirillum* and the decrease in *Roseomonas* phylotypes, changes in the composition of Caulobacteraceae (phylotypes of *Caulobacter*, *Brevundimonas*, *Phenylobacterium*), decrease in *Bosea* phylotypes, and increase in *Afipia*. Gammaproteobacteria were characterized by a decrease in Comamonadaceae (only the genus *Ramlibacter* is represented in the leaf litter) with the presence of genera *Methyloversatilis* and *Achromobacter*. Within the phylum Bacteroidota, the representation of the genus Mucilaginibacter increased, which replaces other genera from Sphingobacteriales (*Parapedobacter*, *Olivibacter*).

For taxonomical comparisons of consortia, we used two approaches - relative abundance data and compositional analysis of balances, which allowed us to detect differences at and below genus level. Our data shows that consortia isolated from enrichment cultures of different substrates are mostly bacterial and differ in the composition of major phylotypes. Each substrate left an imprint on the overall taxonomic composition; however, the composting facilitated growth of specific microbiota, which were classified into different phylotypes, but attributed to common taxonomic groups.

### 2.2. Functional Analysis of Cellulolytic Consortia

Cellulolytic capacities of obtained consortia were accessed by full metagenome sequencing on the ONT platform. Four metagenomes were assembled with comparable metrics (Table 2). The number of contigs varied between 1466–1789 with the total length ranging between 86–108 Gb. Consortium BL6 had the highest total length of contigs and N50 value. Consortium OS2 had the highest number of contigs with the lowest total length.

#### 2.2.1. Metagenome Taxonomy

The taxonomy obtained by Kraken2 coincided with the results of the 16S rRNA gene libraries. Relative abundance could not be compared because Kraken normalizes by the number of contigs, which is very variable in the assembled genomes. Nonetheless, top taxa from consortia were detected both by amplicon libraries and metagenomes (Appendix A), including representatives of Proteobacteria, Actinobacteriota, Firmicutes, Bacteroidota, Verrucomicrobiota, Myxococcota and Planctomycetota. Some of the major representatives were annotated on the species levels: *Aquamicrobium sediminum*, *Shinella granuli*, *Devosia elaelis*, *Asticcacaulis tiandongensis*, *Opitutus terrae*, *Cellulomonas gelida* in OS2 and OS4, *Sporocytophaga myxococcoides* in OS2 and PS4, *Pseudoxanthomonas japonensis* in OS4 and BL6, *Achromobacter denitrificans*, *Ruminiclostridium hungatei* and *Methyloversatilis discipulorum* in BL6, *Youhaiella tibetensis* in PS4.

#### 2.2.2. MAGs

Based on the metagenomic assemblies, 19 (2 from OS2, 5 from OS4, 5 from PS4-5, 6 from BL6) metagenome-assembled genomes (MAGs) were described that meet the following requirements: completeness of more than 90%, contamination of less than 5%. (Appendix A). Around half of the MAGs attributed to the most abundant microorganisms, revealed by amplicon sequencing, such as *Cellulomonas*, *Sporocytophaga*, *Pseudoxanthomonas*, *Devosia*, *Asticaccalulis*, *Opitutus*, which had a great variety of pathways for lignocellulose degradation (Figure 5). There were also MAGS attributed to genera, which presented a smaller component of the consortia: *Dokdonella*, *Luteimonas*, *Afipia*, *Terricaulis*, *Methyloversatilis* and *Magnetospirillum*, which did not have as many genes with CAZy domains. Additionally, these MAGs had genes responsible for microbial interactions: quorum sensing, antimicrobial resistance, and CRISPR-Cas systems. All MAGs show genes associated with the nitrogen, sulfur, and methane metabolic activities, especially those from BL6 consortium.

#### 2.2.3. CAZy Composition

All CAZy categories were distributed similarly across all consortia (Table 3). At the first place were GHs (41.9–47%), then GTs (26.1–30.6%), CBMs (10.2–12%), CEs (9.5–11.1%), AAs (2.6–4.1%), and PLs (1.4–2.6%). Most CAZymes are modular with genes encoding several domains from one or more of the CAZy categories [37]. Thus, we tried to address the composition of CAZy modules in the metagenomes. In our consortia, the most frequently occurring pairs of modules in CAZyme genes belonged to combinations of GH, CBM and GT modules: GH13 + CBM48 (43), GH5 + GH9 (48), GH43 + GH51 (13), GT2 + GT4 (12) (Appendix A).

It has been reported that CAZyme genes, along with transcription factors and transporters, are associated in clusters of several closely spaced genes, the so-called CAZyme gene clusters—CGCs [38]. CGC-finder demonstrated that, in all four consortia, around 32.5% of CAZyme genes were grouped into CGCs, which contained from 2 to 35 genes, with a median of 5 genes. BL6 consortium had the biggest cluster size and proportion of CAZyme genes in it. Most frequently CGCs were found in Rhizobieceae (*Devosia*, *Bosea*, *Shinella*, *Bradyrhizobium*, *Agrobacterium*, *Aquamicrobium*), Xanthomonadaceae (*Pseudoxanthomonas* and *Luteimonas*), and Paenibacillaceae (*Cohnella*) (Appendix A).

In total, we detected 2854 GH genes, which were attributed to approximately 120 different bacterial genera (Appendix A). The highest number of GH genes were found in *Devosia* (Alphaproteobacteria), *Cohnella* (Firmicutes), *Cellulomonas* (Actinobacteriota), *Asticcacaulis* (Alphaproteobacteria), *Pseudoxanthomonas* (Gammaproteobacteria), and *Sporocytophaga* (Bacteroidota), which were detected as major genera by the amplicon sequencing. Most of the other major phylotypes identified at the amplicon sequencing stage were confirmed to have GHs, apart from *Caenimonas* and *Chryseobacterium*, which did not have GHs at all, and Flavobacterium, *Pirellula*, *Terrimonas*, *Dyadobacter*, and *Protochlamydia*, which had singular representatives of GH.

The most widely represented groups of GH families are shown in Figure 6. The functional pattern is similar for all consortia, with the prevailing presence of GH13, GH43, GH23, and GH3 families. However, the opposite situation could be described for CBM-their pattern differs between consortia (Appendix A). Families CBM6, CBM 35, and CBM44 were more characteristic of the consortium PS4, families CBM2 and CBM13, of OS2 and OS4, and families CBM16, CBM32, CBM61, and CBM66-of BL6. The highest proportion value of CBMs in the metagenomes was in BL6 consortium (3.2%), and the lowest in OS2 (1.7%).

Combined 16S and metagenomic data show that most of the major phylotypes are attributed to the potential active cellulolytic bacteria in the isolated microbial consortia.

#### 2.2.4. Gene Set Enrichment Analysis

Gene Set Enrichment Analysis (GSEA) is modeled to be used with a differential expression analysis tool in order to determine whether a predefined set of genes (such as GO term or KEGG pathway) shows statistically significant differences between two biological states. Due to the specifics of our data, we applied GSEA to determine which KEGG categories, not limited to CAZy, are over-represented in metagenomes of our cellulolytic consortia. It was performed pairwise between OS4, PS4, and BL6. After the comparisons, the most characteristic KEGGs of each consortium were selected (Appendix A).

The OS4 consortium was very similar to OS2 and was characterized by the over-representation of ion transport genes (TonB receptor K02014, siderophore transport K02014, K02031), ABC transporters associated with oligosaccharide transport (K02035, K02031), peptidase activity (K00281, K01256, K01354, K00681, K01941), 4-hydroxyphenylpyruvate dioxygenase (K0047), transposase (K07497), exporters (K06147, K18138, K03296), aldehyde dehydrogenase (K00128). The adhesin transporter genes (K21449, K02031) should also be noted, which can be associated with quorum sensing and the high representation of Myxococcota in this consortium.

In the PS4 consortium we detected a higher representation of ABC transporters and permeases, often associated with oligosaccharide transport (K01990, K01992, K02004, K02025, K01998, K01998), ABC exporters (K03296, K03296), ion transport genes (TonB receptor K02014, siderophore transport K02014, heavy metals-K15726, K03455). At the same time, there was an over-representation of genes of energy metabolism for the final stages of fermentation (alcohol dehydrogenase K00001) and respiration (K02274, K01681), genes associated with translation (K01872, K01426, K01876, K01870, K03977), genes associated with cell division (K03654, K02621, K03168, K03654), histidine kinase associated with chemotaxis and cell–cell signaling (K06596, K05874, K07716), serine-threonine protein kinases (K08884, K12132). From the CAZyme genes, GH5, GH9, and GH3 were overrepresented. It is worth noting the presence of genes of a heavy metal export system. To conclude, this consortium is characterized by an over-representation of translation and cell division genes.

Consortium BL6 was enriched by the following categories: oligosaccharide transporters (K02026, K17318, K10548, K08483, K02027, K10119), including permease (K10118, K17319, K02004), ABC exporters (K01990, K06147, K18138), ion transport genes (TonB receptor K02014, siderophore transport-K02015, K02016, K07165), repair polymerases and endonucleases (K02337, K16898, K02337), genes responsible for chemotaxis (K03406, K03406), serine histidine kinase (K03407), arabinose operon regulator (K07720), sporulation regulator (K06297). All in all, in BL6 consortium, the genes for signal transduction proteins, and specific oligosaccharide transporters were overrepresented. There appears to be a definite response to nutrient deficiencies.

According to the GSEA, the main differences in cellulolytic consortia did not show in the CAZyme genes, but mostly in genes of transport and energy systems, and cell division.

#### 2.2.5. Taxonomy Evenness in KEGGs

Classification of the four consortia on the phylum level based on ORFs, GH, CBM, GT, and CGCs revealed taxonomic unevenness between these categories (Figure 7). Proteobacteria phylum was dominant in all four consortia by ORFs, but other phyla have a proportionally higher impact of GHs and CBMs in the metagenomes in comparison to ORFs. In OS2 consortium, they were Actinobacteroidota, Bacteroidota, and Verrucomicrobiota. In OS4—Myxococcota, in PS4—Actinobacteroidota and Bacteroidota, in BL6—Firmicutes.

To look into functional unevenness between consortia on the deeper taxonomic level, we used the inverted Simpson index for KEGG categories on the genus level. KEGGs were filtered by the number of occurrences (>50 in all metagenomes), and only whose values of the inverted Simpson index fell into the 5th percentile were left, 47 KEGGs in total (Appendix A). This cut-off ensured that we were left only with large groups of KEGG categories, which were dispersed unevenly between genera (Appendix A). There was a statistically significant association (generalized linear model with quasi-Poisson family distribution) between the inverted Simpson index and the coefficient of variance of these KEGGs between consortia (Appendix A). Thus, using this approach, we elected KEGGs from different genera from different consortia. Selected KEGGs were the most often associated with the genera *Devosia*, *Cohnella*, *Shinella*, and *Bosea*. Functionally, the genes from these KEGGs are highly specific, and often overlap with those found by GSEA. The smallest evenness is characteristic of ABC transporters and membrane receptors of two-component systems. Of the enzymes associated with the metabolism of carbohydrates, it is worth highlighting the aldouronate-utilization operon components characteristic of the Paenibacillaceae family, including the regulators of oligosaccharide metabolism YesN/YesM (K07718, K07720, K03534), permeases of the aldouronate transport system (K17319, K17318, K17320). In addition, coinciding with GSEA, inverted Simpson index selected groups of oligosaccharide transporters (K10111, K10440, K01473, K15771) and exporters, most often characterized as transporters of metal ions and peptides (K09969, K10823, K09969). Among these enzymes, some groups were associated not with the metabolism of carbohydrates, but rather with the formation of a microbial community, such as quorum sensing genes (K02031, K02033). It should also be noted that CAZymes were not selected by this approach, which, together with the results of the analysis of MAGs and GSEA, allows us to say that these enzymes are not taxon-specific for the cellulolytic consortia that we obtained. Characteristically, these functional groups (ABC transporters and two-component systems), revealed by invSimpson analysis and GSEA, overlap with the functional groups associated with CAZy in the CGCs.

## 3. Discussion

### 3.1. Diversity of Members of the Cellulolytic Consortia

There is evidence that only a few microorganisms are sufficient for the destruction of lignocellulose [39]. However, the diversity of microorganisms in selected consortia from our experiment was, according to the results of the amplicon sequencing, about a hundred phylotypes. This can be explained by both methodological and biological factors. In our experiment we aimed to dilute composting lignocellulose substrate to the extent that only active cellulolytic organisms would be able to survive on a medium with filter paper as a sole carbon source. Indeed, major phylotypes from all consortia we selected attributed to genera, most commonly characterized as mesophilic aerobic and facultative anaerobic cellulose degraders from soil and sediments: *Sporocytophaga myxococcoides* [40,41], *Cellulomonas* [42,43,44,45]. *Flavobacterium* [46,47], *Pseudoxanthomonas* [48,49], *Asticcacaulis* [50,51], *Paenibacillus* [52,53], and *Devosia* [54]. However, each consortium contained specific microorganisms, most likely not associated with the direct decomposition of the lignocellulosic complex. *Caenimonas*, a soil mesophile aerobe [55,56], was one of the components of OS4 consortia, but it did not have GHs according to metagenome data. Another striking example of such a component was *Bdellovibrionota*-predatory microorganism [57], found as a minor component in all consortia. Interestingly, Myxococcota, which was represented in the OS4 consortium, had a lot of GHs, but they are also characterized as predatory [58]. This could explain their prevalence in the consortium selected from the oat straw, composted for a longer period. The high diversity of the bacterial component can also be associated with the almost complete absence of eukaryotes in the dataset [59]. Despite some representation of the fungal community in amplicon ITS sequencing, most of the data were not identified in known databases. Three representatives belong to Ciliophora, which also has bacterivorous species [60]; thus, they also add to the non-cellulolytic part of the selected microbial consortia.

The relationship between microorganisms in lignocellulolytic communities can be expressed not only through competition for the substrate. As an example, lignocellulosic complexes in the process of degradation can themselves exhibit different antimicrobial activity due to decomposition into specific secondary metabolites that are toxic to a part of the microflora [61]. In addition, during cellulose decomposition. biofilm formation often occurs, especially under conditions of reduced aeration [62]. According to the amplicon sequencing, we have shown that the leaf litter consortium BL6 contains both aerobes (*Achromobacter*) [63], microaerophiles (*Methyloversatilis*, *Magnetospirillum*) [64,65], and strict anaerobes (*Ruminiclostridium*, *Anaerocolumna*, *Herbinix*) [66,67,68], which can be a sign of biofilm formation in this consortium, since they are often formed in nature during the breakdown of lignocellulose [69]. In the consortium OS4, we observed *Flavobacterium* as the major component, which also was reported to be able to form biofilms [70].

Taxonomic differences between consortia were revealed both in the high and low taxonomic resolution. On the phylum level, all consortia consisted predominantly of Proteobacteria, Actinobacteriota, and Bacteroidota, but each consortium had an individual imprint of minor phyla. For OS2, it was Verrucomicrobia, OS4–Myxococcota, Planctomycetota, and Firmicutes. Analysis of taxonomical composition by PhILR package showed that substrate specificity is associated with transitions at the taxonomic level below the genus. What this means is that different microbial communities from different natural substrates demonstrated the process of convergence, where similar, but not identical bacterial genera were given growth advantage. These results are enforced by metagenome data, where we can look more deeply into differences at the species level.

### 3.2. Functional Metagenome of the Cellulolytic Consortia

Four isolated consortia had similar profiles of GHs. Most of the GH families detected in the consortia were associated with lignocellulose decomposition, with some exceptions. The most abundant GH group in the consortia belonged to the family GH13, which predominantly consists of enzymes acting on α-glycosidic bonds, thus not partaking in lignocellulose composition [71]. Another example of a major GH group not participating in cellulose degradation is family GH23, consisting predominantly of lysozymes and peptidoglycan lyases [72], probably contributing to interaction with other members of the consortium. However, there is evidence that this family contains some unconventional genes, involved in cellulose decomposition [73]. The most abundant cellulose-related families detected in consortia (GH43, GH3, GH2, GH16, and GH1), were connected to genes encoding enzymes from the last stage of cellulose and hemicellulose decomposition: glucosidases, xylosidases, galactosidases, etc. [74]. GH families, encoding enzymes from the first stages of cellulose and hemicellulose decomposition (GH5, GH51, GH10, GH9 and GH6), were present, but less abundant [75]. Thus, isolated consortia demonstrated the later stages of lignocellulolytic activity alongside defense functions and utilization of simple sugars.

Despite the high number of CAZy modules in the individual genomes and the metagenomes as a whole, their quantities did not differ significantly in the consortia, but actual differences were detected for the proteins of the transport systems and membrane regulatory domains. Genes, characteristic of a certain consortium, were transport proteins, genes associated with two-component systems (receptors) and housekeeping genes (replication, repair, chaperones). At the same time, it is difficult to interpret what role different transport proteins play. Most often, we cannot divide this group functionally because it contains both ABC transporters associated with ion transport and peptide export. It is known that the presence of copper transport, which is a limiting macronutrient for microbial communities, from the cell is associated with ligninase activity (aldehyde dehydrogenase), [76]. It should not be ruled out that ABC transporters are an important part of protective enzymes, which in turn can be associated with the important role of microbial selection in the community [77]. It should be noted that the presence of exporters can be associated both with the export of catalases and with the protective function associated with microbial–microbial interactions. This is also supported by the fact that all metagenomes of the consortia have antibiotic resistance genes (kanamycin, sulfisoxazole), CRISP systems, receptors associated with quorum sensing and toxin/antitoxin systems. Of the genes that may play a role in the formation of biofilms in the studied communities, adhesins and spermidine synthesis genes are worth noting, which is in good agreement with the data obtained based on amplicon sequencing data [78]. Among the transport systems, specific cassettes associated with the transport of oligosaccharides were found. This agrees with the fact that around half of detected CGCs contained transport proteins.

In our consortia, we detected widely known cellulolytic bacteria, such as *Sporocytophaga* and *Cellulomonas*. At the same time, we detected the genomes of cellulolytic microorganisms, which are not widely represented in the literature devoted to biotechnology, as a major component of the consortia. For example, representatives of Verrucomicrobia are very widely represented in the soil, but many of them are uncultivated [79]. The genome of the representative of Verrucomicrobia (*Opitutus*) collected by us contained a high representation of glycoside hydrolases. *Magnetospirillum* isolated from the leaf litter consortium is also an unexpected representative of cellulolytic communities. In addition, the presence of microorganisms associated with methylotrophy, *Methyloversatilis discipulorum* [80], was shown in the leaf litter consortium. In the article devoted to the early decomposition of oak leaves, the presence of the dominance of *Methylosinus trichosporium*, which also has methylotrophic activity, was shown in the early stages of decomposition [80]. *Magnetospirillum* has a similar metabolic profile [81]. Methylotrophy is a specific feature of the microbial process of lignocellulose degradation, in contrast to the fungal one. The methylotrophic component found in the litter consortium is of considerable interest to biotechnologies because it allows you to increase the yield of an organic product [82].

## 4. Materials and Methods

Oat straw, pine sawdust, and birch leaf litter were used to isolate minimal cellulolytic consortia. The experiment was laid out in October 2018. Three replicate samples of each substrate weighing 60 g were moistened up to 60% of the total capacity and mixed with mineral fertilizer (NPK elements in 16:16:16 ratio) to a concentration of 10 mg/g of the substrate. All substrates and fertilizer were non-sterile and did not have any additional microbial inoculum. Fertilizer was added to facilitate decomposition by removing macronutrient deficiency, characteristic of C-rich biomass. The mixture of each substrate in three replicates was composted in plastic containers at 28 C for 6 months. At two, four, and six months it was remixed, and each substrate was sampled for the enrichment test. Each sample was processed with a series of 10-fold dilutions in sterile water, which were inoculated into 5 mL of Hutchinson liquid medium (g/L: K_2_HPO_4_—1; NaCl—0.1; CaCl_2_—0.1; FeCl_3_—0.01; MgSO_4_*7H_2_O—0.3; NaNO_3_—2.5) with ashless filter paper as the only carbon source [83]. We looked for microbial consortia grown from the largest dilutions, which preserved the ability to macerate the filter paper. After the experiment, only four stable microbial consortia with reproducible phenotypes and cellulolytic activity after several months of transfers were selected for the 16S rRNA libraries and full metagenome sequencing.

For DNA extraction fresh microbial consortia were incubated in tubes for two weeks in a new portion of Hutchinson liquid medium. In total, six replicate cultures for each of the four consortia were acquired. After the incubation period, the contents of the tubes were centrifuged, and the supernatant was removed. The cell debris with the remaining filter paper was resuspended in SL1 buffer from NucleoSpin^®^ Soil Kit (Macherey-Nagel GmbH& Co. KG, Düren, Germany) and ground using a Precellys 24 homogenizer (Bertin Technologies, Saint-Quentin en Yveline, France). This mixture was used for DNA extraction according to the manufacturer’s recommendations. Construction and sequencing of the 16S rRNA amplicon libraries were performed on the Illumina MiSeq platform (Illumina, Inc., San Diego, CA, USA) as described previously [84].

Amplicon libraries were processed using the DADA2 pipeline [85] in the R software environment [86]. Taxonomic identification was carried out using the Silva 138 database [87] for 16S rRNA gene sequences and the Unite database [88] for ITS sequences. The phylogenetic tree was constructed using the IQ-TREE 2.1.2 program [89]. Further processing was carried out using the phyloseq [90] and ampvis2 [91] packages. Alpha diversity was accessed by three indexes: Observed, Shannon [92], and inverted Simpson [93], with the significance of mean differences between them calculated by the Mann–Whitney test [94]. Beta diversity was accessed by PCoA [95] with the Bray–Curtis distance matrix [96]. A composite data analysis (phylogenetic isometric log-ratio) based on the PhILR package [97] was used to search for distinct groups of microorganisms.

For the metagenome sequencing, all six DNA samples were mixed for each consortium. The resulting four mixtures were used for the library construction with the ligation sequencing kit (SQK-LSK110) using genomic DNA by ligation protocol (SQK-LSK110) and sequencing on the MinION platform with the Flow Cell R9.4.1 (Oxford Nanopore Technologies, Oxford, UK). The initial processing of the full metagenomic reads was conducted with the Guppy 3.6.1 neural network model [98]. The assembly was performed in the metaFlye 2.8 program [99], additional cleaning of the resulting contigs was carried out using algorithms included in Racon [100] and medaka [101].

Taxonomic classification of contig-assembled metagenome sequence data was performed by Kraken2 [102]. Prokaryotic contigs were assigned using the pre-built database based on GTDB 202 [103], eukaryotic-on the pre-built custom database maxikraken2 1903 based on RefSeq NCBI [104]. Visualization of relative abundance was performed in Krona [105].

The annotation of Carbohydrate-Active Enzymes in the assembled contigs was obtained using dbCAN [106]. An ORF match was considered if two tools from DIAMOND [107], HMMER or Hotpep [108] annotated it. The search of CAZyme gene clusters (CGCs) in assembled contigs was performed by the CGC-Finder [109]. The rest of the genes were annotated using EggNOG-mapper (v.2.1.6) pipeline in ultra-sensitive DIAMOND mode with the --framefix flag [110,111]. Gene overrepresentation analysis (GSEA) was performed using the ClusterProfiler (v 1.3.1) package [112] using EggNOG-mapper with KEGG and COG functional classifications with Fisher test [113] and FDR correction method [114] for multiple comparisons. Taxonomic unevenness of KEGGs was calculated in vegan [115].

High-quality MAGs were isolated from the obtained assemblies using the MetaBat2 program [116] using the following parameters: over 90% completeness and less than 5% heterogeneity as defined by the CheckM program [117]. MAGs were taxonomically annotated with the GTDB-Tk toolkit using the HMMER, pplacer, FastANI, and FastTree algorithms based on the GTDB 202 database [118]. Functional annotation was carried out using the DRAM pipeline using the UniRef90, PFAM, dbCAN, and MEROPS databases [119]. The code is available at https://github.com/crabron/cellulolytic_consortia, accessed on 12 August 2022.

## 5. Conclusions

In our work we isolated four stable cellulolytic consortia from various natural lignocellulose substrates by making enrichment cultures from diluted compost with the filter paper as a sole carbon source. Each consortium was a complex microbial community with distinctive phenotypic features. At the core of every consortium were cellulolytic bacteria from various genera; however, in addition to them, auxiliary components were present, such as biofilm inhabitants, predators, or methylotrophs. Being taxonomically different, the composition of CAzyme genes in the consortia coincided. The main differences between consortia were detected in the taxon-specific transport and regulatory genes. The similar conditions of selection and cultivation functionally brought together taxonomically different consortia; however, each substrate left an individual imprint on their composition.

## Figures and Tables

**Figure 1 ijms-23-10779-f001:**
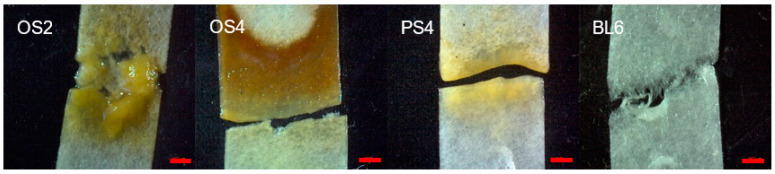
Maceration of the filter paper by each microbial consortium. OS2—oat straw, 2-month composting period; OS4—oat straw, 4 month; PS4—pine sawdust, 4 month; BL6—birch leaf litter, 6 month. The red scale bar is 2 mm.

**Figure 2 ijms-23-10779-f002:**
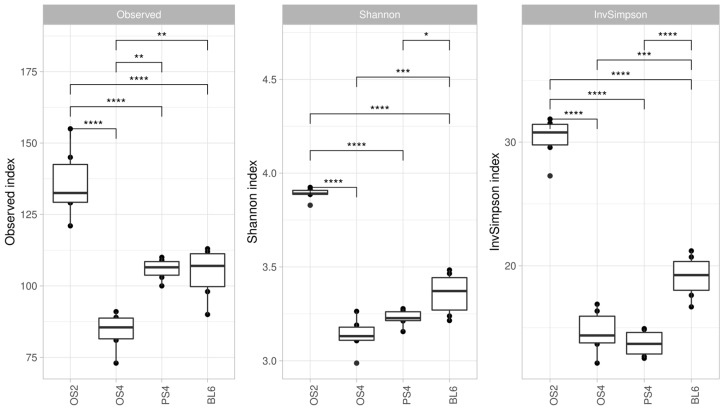
Alpha-diversity of four consortia (OS2, OS4, SD4, BL6) accessed by 3 indices—Observed, Shannon, InvSimpson, based on 16S rRNA gene libraries. Significant differences assessed by the Mann–Whitney test: (*) *p*-value ≤ 0.05; (**) *p*-value ≤ 0.01; (***) *p*-value ≤ 0.001; (****) *p*-value ≤ 0.0001.

**Figure 3 ijms-23-10779-f003:**
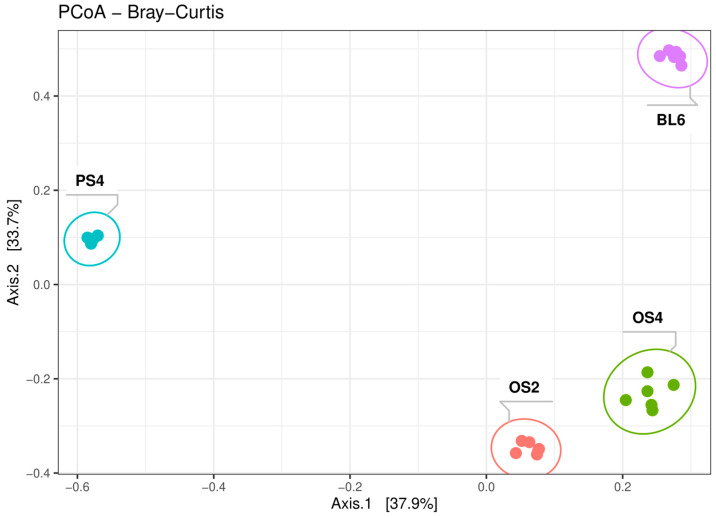
PCoA plot of the beta-diversity of the four cellulolytic consortia (OS2, OS4, PS4, BL6) accessed with Bray–Curtis, based on 16S rRNA gene amplicon libraries.

**Figure 4 ijms-23-10779-f004:**
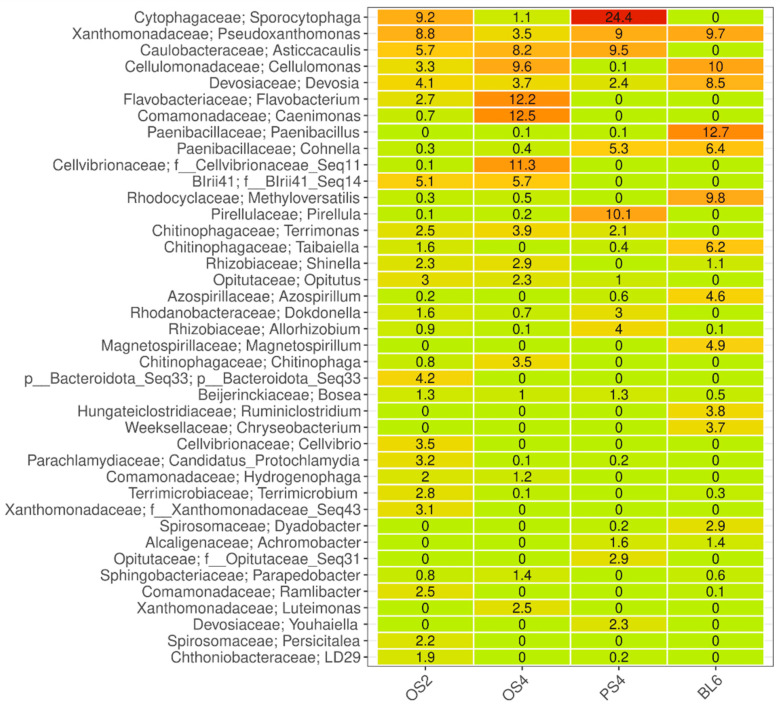
Heatmap of the most abundant bacterial phylotypes on the genus level in the four cellulolytic consortia (OS2, OS4, PS4, BL6). Relative abundance given in % of the read count of each consortium, red for maximal, and green for minimal values.

**Figure 5 ijms-23-10779-f005:**
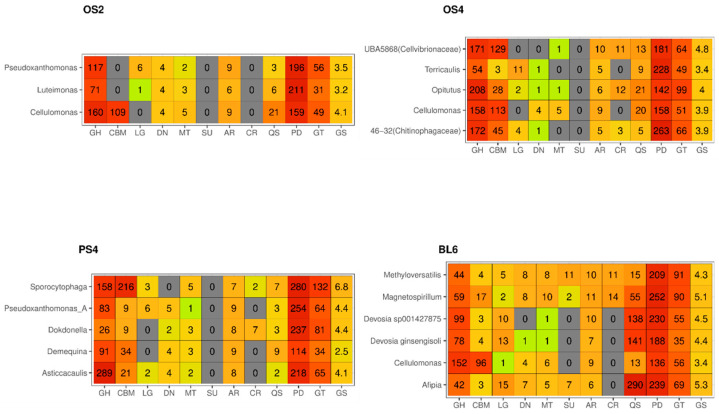
MAGs from the four consortia and representation of functional genes related to substrate decomposition and microbial interactions. Heatmaps are based on the DRAM annotation. Increased hit value relative to the dbCAN/EggNOG annotation caused by using a greedier algorithm. GH-Glycoside Hydrolyses (DRAM); CBM-Carbohydrate-Binding Modules (DRAM); LG—Ligninases (eLignin Database pathways, Metacyc, EC); DN—Denitrification (DRAM); MT—C1-methane (DRAM); SU—Sulfur metabolism-sulfite reductase, thiosulfate reductase, thiosulfate oxidation by SOX complex, thiosulfate ≥ sulfate (DRAM); AR—Antibiotic Resistance (DRAM); CR—CRISPR (DRAM); QS—quorum sensing (KEGGs from ko02024 path); PD—Peptidases (DRAM); GT—GlycosylTransferases (DRAM); GS—Genome Size in Mb.

**Figure 6 ijms-23-10779-f006:**
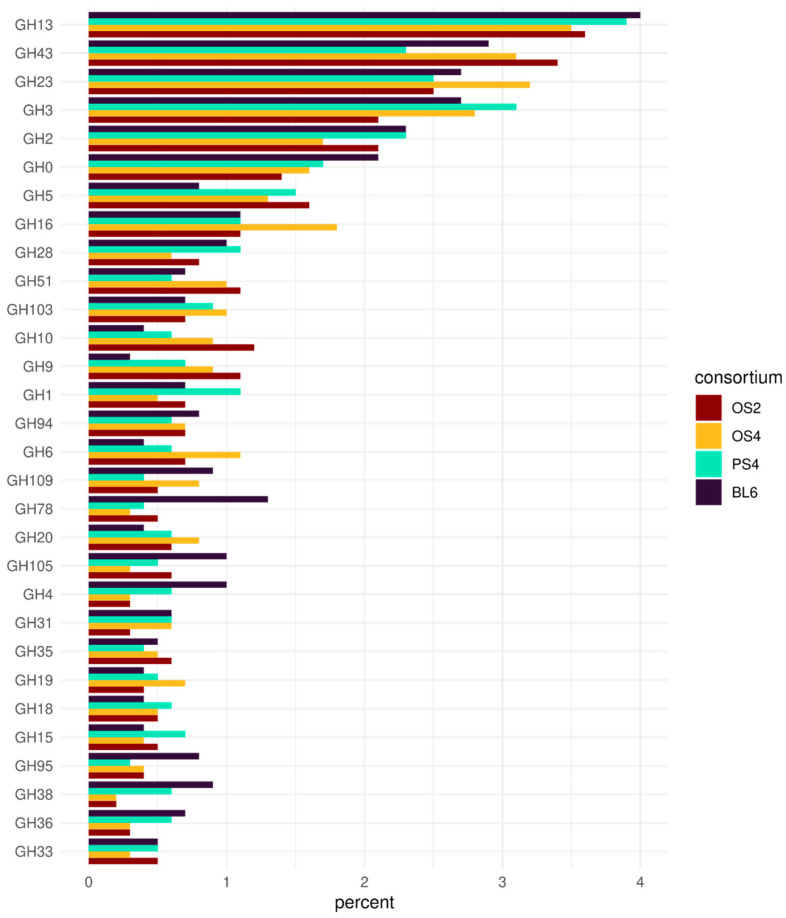
GH families in the four cellulolytic consortia (OS2, OS4, PS4, BL6).

**Figure 7 ijms-23-10779-f007:**
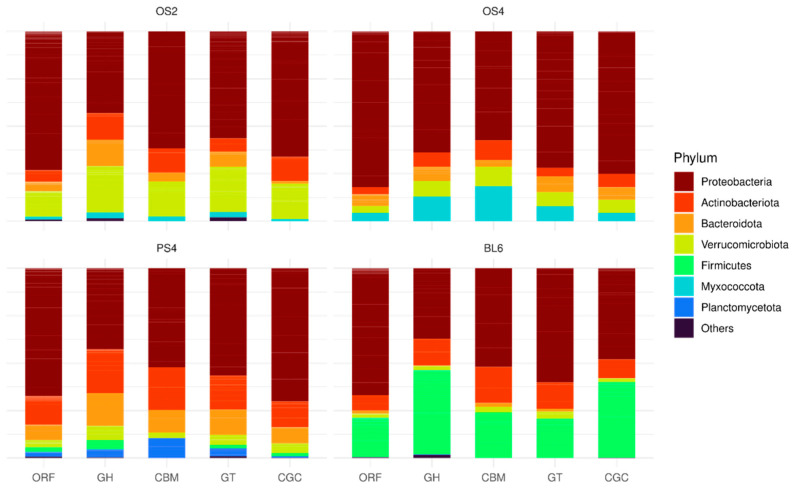
Taxonomic composition on the phylum level in the four cellulolytic consortia (OS2, OS4, PS4, BL6), based on ORFs, GHs, CBMs, GTs, and CGCs annotations.

**Table 1 ijms-23-10779-t001:** Characteristics of isolated microbial consortia.

Consortia ID	Substrate	Composting Time, Months	Substrate Dilution	Color of Consortia	Gas Formation
OS2	oat straw	2	10^6^	yellow	no
OS4	oat straw	4	10^6^	brown	yes
PS4	pine sawdust	4	10^3^	yellow	no
BL6	birch leaf litter	6	10^4^	transparent	yes

**Table 2 ijms-23-10779-t002:** Characteristics of assembled metagenomes of isolated microbial consortia.

Consortium ID	Contigs	Longest Contig, bp	Total Contigs Length	N50
OS2	1879	1,582,432	86,161,676	72,582
OS4	1612	5,307,558	96,558,959	131,290
PS4	1466	2,705,151	87,318,114	112,897
BL6	1748	3,656,276	108,520,437	186,463

**Table 3 ijms-23-10779-t003:** CAZy module composition in metagenomes of isolated microbial consortia, % of CAZy count. AAs—Auxiliary Activities, CBMs—Carbohydrate-Binding Modules, CEs—Carbohydrate Esterases, GHs—Glycoside Hydrolases, GTs—Glycosyl Transferases, PLs—Polysaccharide Lyases.

Consortium ID	AAs	CBMs	CEs	GHs	GTs	PLs
OS2	4.1	10.8	10.7	43.1	29.2	2.2
OS4	3.7	11.3	11.1	41.9	30.6	1.4
PS4	2.6	10.2	11.1	44.3	30.0	1.8
BL6	2.7	12.0	9.5	47.0	26.1	2.6

## Data Availability

All acquired data is available at the BioProject Database (https://www.ncbi.nlm.nih.gov/bioproject/, accessed on 12 August 2022) via Accession: PRJNA844230.

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
