# Peer review of "The Structure of Stable Cellulolytic Consortia Isolated from Natural Lignocellulosic Substrates"

_ijms, 2022, doi:10.3390/ijms231810779_

Round 1
Reviewer 1 Report
Overall, this manuscript is very readable, the introduction provides comprehensive background information and clearly states the research gaps. Methods are well documented, and the results are well described. I especially appreciate the fancy figures.
The authors just need to pay attention to the details, for example:
Table 3, keep the constant significant figure in the entire table. e.g., CBMs for BL6 should be 12,0 instead of 12.
Author Response
Overall, this manuscript is very readable, the introduction provides comprehensive background information and clearly states the research gaps. Methods are well documented, and the results are well described. I especially appreciate the fancy figures.
Or team really appreciates your overall positive review. This theme is quite new to us, and we are glad we were able to add something valuable to the field.
The authors just need to pay attention to the details, for example:
Table 3, keep the constant significant figure in the entire table. e.g., CBMs for BL6 should be 12,0 instead of 12.
Thank you for this notice. I added missing significant figures. Additionally, I changed commas to dots in decimals to correspond with the rest of the manuscript. Some overlooked typos were also corrected.
Reviewer 2 Report
The authors used plant biomass derived from various natural products as substrates to create four types of consortia. The consortia were characterized by genomic analysis. This will be of great value for the future use of biomass with these consortia.
I have some suggestions for improvement.
(1) What does consortium stability refer to, reproducibility? From which results, figures, tables, etc., can its stability be determined?
(2) In the Discussion, the authors mentioned that BL6 and OS4 potentially have the ability to form a biofilm. Couldn’t they form a biofilm this time?
(3) Figure S1: In OS, the consortia created at 2 and 4 months are different, what were the results for the other samples (SD2, SD6, BL2, BL4, OS6)?
(4) Figure S4: There seems to be no mention of this figure in the text.
Author Response
The authors used plant biomass derived from various natural products as substrates to create four types of consortia. The consortia were characterized by genomic analysis. This will be of great value for the future use of biomass with these consortia.
Thank you for your contribution into reviewing this manuscript, it is very valuable to us. We tried to incorporate a lot of analyses from our research into one comprehensible paper, and we are glad if this somehow succeeded.
I have some suggestions for improvement.
- What does consortium stability refer to, reproducibility? From which results, figures, tables, etc., can its stability be determined?
Yes, you are right, by stability we meant reproducibility. We state that our consortia were reproducible after several months of transfers, but unfortunately, I don’t think that we have figures illustrating this process. Figures presented in the main text and the supplement demonstrate cultures used for the genetic analyses, not the original cultures from biomass dilution. I tried to specify this in the materials and the results.
- In the Discussion, the authors mentioned that BL6 and OS4 potentially have the ability to form a biofilm. Couldn’t they form a biofilm this time?
Maybe we were unclear, but we implied that some of our consortia contained microorganisms, which are capable of biofilm formation. Additionally, in the literature cellulolytic consortia were shown to be able to form a biofilm. Thus, we assume that there may be biofilms in our consortia, but at this time we do not prove it. Maybe there were biofilms, maybe not, but there is a high possibility that our consortia formed biofilms. I tried to add clarification to the phrase in the discission.
- Figure S1: In OS, the consortia created at 2 and 4 months are different, what were the results for the other samples (SD2, SD6, BL2, BL4, OS6)?
Truly, in our research we cover only some stages of the biomass composting experiment. Initially we did enrichment test for all samples, but in the end, we selected only those which resulted in reproducible consortia, thus only four of them.
- Figure S4: There seems to be no mention of this figure in the text.
Yes, this is a typo, Figure S5 was mentioned twice, the first time it should have been figure F4. I corrected it. Plus I added another supplemental figure, thus all numeration was changed